# The Magnetocaloric Behaviors of Gd-based Microwire Arrays with Different Curie Temperatures

**Hongxian Shen** [1,2], **Lin Luo** [1], **Hillary Belliveau** [2], **Sida Jiang** [3,*], **Jingshun Liu** [4], **Lunyong Zhang** [1], **Yongjiang Huang** [1], **Jianfei Sun** [1,*] and **Manh-Huong Phan** [2,*]

1  School of Materials Science and Engineering, Harbin Institute of Technology, Harbin 150001, China
2  Department of Physics, University of South Florida, Tampa, FL 33620, USA
3  Space Environment Simulation Research Infrastructure, Harbin Institute of Technology, Harbin 150001, China
4  School of Materials Science and Engineering, Inner Mongolia University of Technology, No. 49 Aimin Street, Hohhot 010051, China
*  Correspondence: jiangsida@hit.edu.cn (S.J.); jfsun@hit.edu.cn (J.S.); phanm@usf.edu (M.-H.P.)

**Abstract:** The desirable table-like magnetocaloric effect (MCE) was obtained by designing a new magnetic bed, which comprises three kinds of Gd-based microwire arrays with different Curie temperatures ($T_C$). The $T_C$ interval among these wires is ~10 K. This new magnetic bed shows a smooth ferromagnetic to paramagnetic transition at ~100 K. In addition, a table-like magnetic entropy change ($\Delta S_M$) was obtained, ranging from ~92 K to ~107 K, with a maximum entropy change ($-\Delta S_M{}^{max}$) of 9.42 J/kgK for a field change ($\mu_0 \Delta H$) of 5 T. Notably, the calculated results of $-\Delta S_M(T)$ corresponded to the experimental data for $\mu_0 \Delta H = 5$ T, suggesting that a microwire array-based magnetic bed with desirable magnetocaloric response can be designed. In addition, it was shown that a larger table-like temperature range and cooling efficiency can be achieved by increasing the interval of $T_C$ among microwire arrays. These important findings indicate that the newly designed magnetic bed is very promising for active magnetic cooling technology.

**Keywords:** table-like; magnetocaloric effect; microwire arrays





## 1. Introduction

Magnetic refrigeration is a promising alternative to conventional refrigeration techniques, owing to its higher cooling efficiency and environmental friendliness [1,2]. A magnetic material with excellent magnetocaloric effect (MCE) is employed as a core component in a magnetic cooling device. Commonly, the magnetocaloric materials show a maximal magnetic entropy change ($\Delta S_M$) around a magnetic ordering transition temperature, which is of first-order magnetic transition (FOMT) [3] or of second-order magnetic transition (SOMT) [4]. The temperature range within which the maximum magnetic entropy change ($-\Delta S_{Mmax}$) is observed is often considered as the working temperature range for magnetic refrigeration. Obviously, the MCE within the working temperature range is usually inhomogeneous, which is more serious for FOMT materials with a narrow transition temperature range [3].

A regenerative Ericsson cycle is theoretically considered to be ideal for magnetic refrigeration applications above 20 K, due to its high working efficiency and broad temperature range [5]. The peak shape of a MCE or $\Delta S_M(T)$ curve is a limiting factor of the operating temperature of these types of magnetic materials. Therefore, magnetocaloric materials with table-like magnetic entropy changes spanning the whole working temperature range are ideal refrigerants for use in cooling systems with an Ericsson cycle. Table-like MCE features were observed in multiphase structures [6,7] or composites. The different phases usually show different values of $T_C$ and $\Delta S_M$, so the MCE of the multiphase alloy is a combined contribution of these respective phases. The multiphases could be formed during the fabrication process [5] or in situ synthesis through thermal treatment methods. However, it is difficult to control the volume fraction of each individual phase or phases with required

performances. An alternative way to achieve the table-like MCE is to design a composite by directly utilizing material components with different required MCE properties [8,9], such as the fabrication of multi-layered structures composed of different magnetic ribbons [10,11], or sintering magnetic materials with different compositions [12,13].

Clearly, the large size of raw alloys causes the resulting composite to show macroscopic inhomogeneity of compositions and performances. Reducing the size of such raw materials is an effective way to solve the raised problem. Magnetocaloric wires with a micro-size are the ideal candidates for designing table-like MCE composites. In addition, theoretical calculations suggest that a magnetic refrigerator using microwires as refrigerants could yield a high operating frequency [14]. Additionally, the micro-size of wires yields a high heat exchange efficiency, which is beneficial for the cooling system [15,16]. In recent years, excellent MCEs have been reported in Gd-based microwires, which are also superior to their bulk and ribbon counterparts [17–29].

In this work, we have designed a new magnetic bed composed of three Gd-Al-Co melt-extracted microwire arrays with different Curie temperatures for achieving a table-like MCE. It is shown that the MCE and cooling efficiency can be tuned by selecting microwires with appropriate magnetic entropy changes and Curie temperatures.

## 2. Materials and Methods

The Gd-Al-Co microwires used for the design of a magnetic bed were fabricated by the melt-extraction method and the MCE performance can be found in our previous reports [19,20,30]. The required information for these three types of microwires is displayed in Table 1. The Gd-Al-Co microwires exhibit amorphous structures with surrounding maximum magnetic entropy changes but different $T_C$ values with an interval of ~10 K. Figure 1a shows a SEM image of the melt-extracted Gd-Al-Co microwires with a diameter of ~30 μm. The microwires show excellent adaptation of morphology and structure, so a magnetic bed with different porous structures can easily be designed by arranging these microwires. We have designed a magnetic bed by using microwires with equal weight fractions of $Gd_{50}Al_{30}Co_{20}$, $Gd_{50}Al_{25}Co_{25}$, and $Gd_{60}Al_{20}Co_{20}$ compositions, as shown in Figure 1b. A sample holder with a length of ~3mm and an inner diameter of ~1mm was used to hold the wires. The wires with different compositions were arranged inside the holder, in which the axial direction of the wires was parallel to the axial direction of the sample holder.

The morphology of the as-prepared Gd-Al-Co microwires was obtained on a field emission scanning electron microscope (SEM-Helios Nanolab600i, FEI, Hillsboro, OR, USA). The MCE performances of the designed magnetic arrays bed were measured on a commercial Physical Property Measurement System (PPMS-16T, Quantum Design, San Diego, CA, USA) with a magnetic field of up to 5 T.

**Table 1.** The MCE properties of three melt-extracted Gd-Al-Co microwires at field change of 5 T.

| Composition | $T_C$ (K) | $-\Delta S_M$ /(Jkg$^{-1}$K$^{-1}$) | $RC$ (J/kg$^{-1}$) | RCP (J/kg) | Ref. |
|---|---|---|---|---|---|
| $Gd_{50}Al_{30}Co_{20}$ | 86 | 10.09 | 672 | 861 | Present work |
| $Gd_{50}Al_{25}Co_{25}$ | 97 | 10.30 | 622 | 833 | [31] |
| $Gd_{60}Al_{20}Co_{20}$ | 109 | 10.11 | 681 | 915 | [19] |

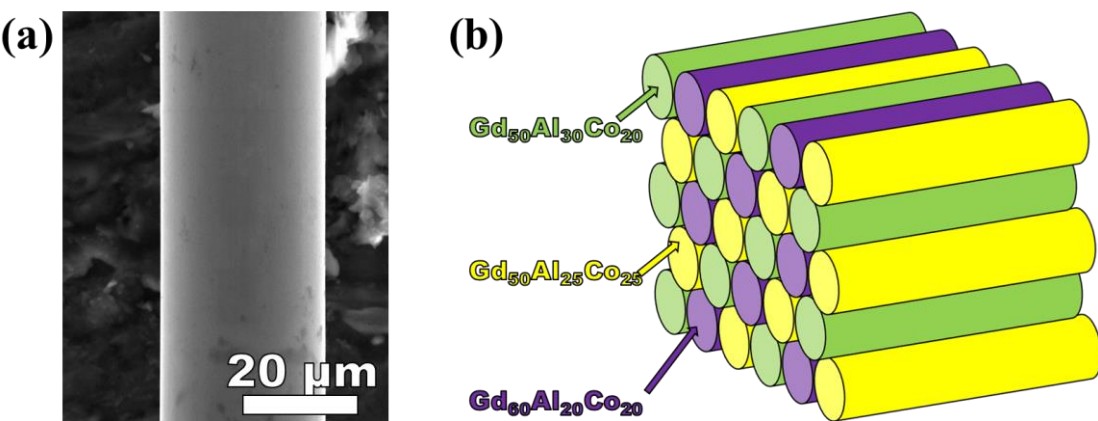

**Figure 1.** (**a**) SEM image of the melt-extracted Gd-Al-Co microwires. (**b**) The magnetic arrays bed designed by using three kinds of microwires.

The magnetic entropy changes ($\Delta S_M$) at different temperatures and for magnetic field changes in the microwire array-based magnetic bed were calculated through a series of isothermal magnetization (*M-H*) curves and Maxwell equation [17]:

$$\Delta S_M = \mu_0 \int_0^{H_{max}} \left( \frac{\partial M}{\partial T} \right) dH \qquad (1)$$

where *M*, *H*, and *T* are the magnetization, magnetic field, and temperature of the designed magnetic bed, respectively. The adiabatic temperature rise ($\Delta T_{ad}$) is another direct way to evaluate the MCE performance of a magnetic refrigerant, which can be calculated as [5]:

$$\Delta T_{ad} = - \int_0^H \frac{T}{C_{P,H}} \left( \frac{\partial M}{\partial T} \right)_H dH = - \frac{T}{C_P(T,H)} \Delta S_m(T,H) \qquad (2)$$

where $C_P$ is the heat capacity. In addition, the cooling efficiency is also important to evaluate the applicability of a magnetic material in the cooling system, which is defined as refrigerant capacity (*RC*) [18]:

$$RC = \int_{T_{cold}}^{T_{hot}} -\Delta S_M(T) dT \qquad (3)$$

where $T_{cold}$ and $T_{hot}$ are the temperatures at the full width at half-maximum (FWHM) of the $\Delta S_M(T)$ curves.

## 3. Results and Discussion

### 3.1. Magnetocaloric Performance

The temperature dependence of magnetization (*M-T*) over a temperature range of 20–200 K was measured in an applied field of 200 Oe for evaluating the magnetic transition temperatures of the designed microwire arrays. The direction of the applied magnetic field is along the axial direction of the microwires. The results corresponding to the *M-T* curves for three kinds of wires are displayed in Figure 2a,b. It is worth noting that the *M-T* curve of the designed microwire arrays-based bed shows a smooth ferromagnetic-paramagnetic (FM-PM) transition with only one extreme point in the corresponding differential curve (*dM/dT* vs. *T*, inset of Figure 2a). That means the designed magnetic bed shows only one $T_C$ at ~100 K, which exhibits a similar tendency with the wire components instead of three $T_C$ temperatures. The above results clearly show the homogeneity in the structures and performances of the designed magnetic bed.

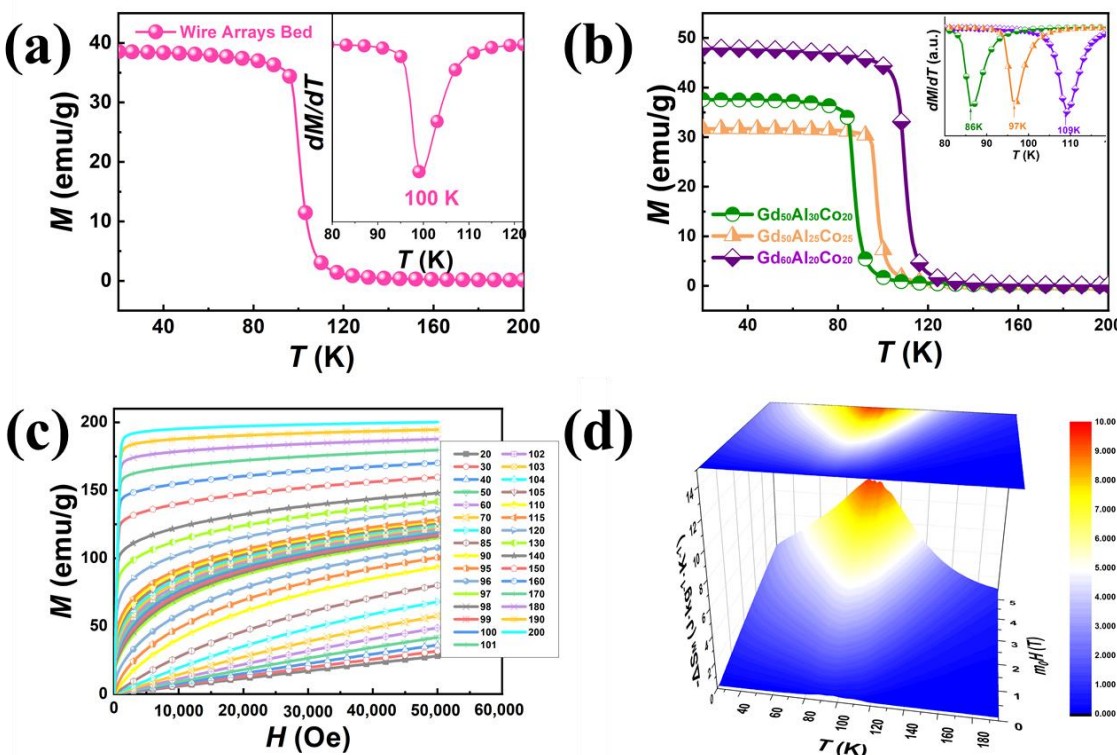

**Figure 2.** The temperature-dependent magnetization $M(T)$ dependence curves for (**a**) The designed bed and (**b**) Three kinds of Gd-Al-Co microwires at an applied field of 200 Oe. (**c**) The isothermal magnetization curves of the designed bed. (**d**) The calculated 3D-plot magnetic entropy changes at different temperatures and field changes.

The *M-H* curves of the designed microwire arrays bed were measured at temperatures ranging from 20 to 200 K, with an interval of 10 K that decreases to 5 K and 1 K near the transition temperatures. The applied field change ($\mu_0\Delta H$) is 5 T, and the results are replotted in Figure 2c. Then, the temperature dependence of magnetic entropy change for different field changes of the designed bed are calculated based on Equation (1) and plotted in three-dimensional form in Figure 2d. The maximum entropy change ($-\Delta S_M^{max}$) at ~100 K with $\mu_0\Delta H$ = 5 T is ~9.42 J/kgK, whose value is slightly lower than those of ~10.09 J/kgK, ~10.3 J/kgK and ~10.11 J/kgK for all its wire components $Gd_{50}Al_{30}Co_{20}$, $Gd_{55}Al_{25}Co_{25}$ and $Gd_{60}Al_{20}Co_{20}$, respectively [19,31]. The decrease of $-\Delta S_M^{max}$ in magnitude for the designed bed is resulted from the $T_C$ interval among the Gd-Al-Co wires. In addition, the peak value of the adiabatic temperature rise ($\Delta T_{ad}$) for the designed bed is calculated to be ~3.7 K at $\mu_0\Delta H$ = 5 T based on Equation (2). For $\mu_0\Delta H$ = 2 T, which is the maximum field that can be provided by permanent magnets, the peak value of $\Delta T_{ad}$ for the designed bed is ~1.74 K. The large $\Delta T_{ad}$ indicates that the microwire array-based bed is a promising candidate for use in active magnetic coolers. Notably, the designed microwire arrays-based bed exhibits a table-like MCE behavior and the width of the table-like range is ~15 K, from ~92K to 107 K, for $\mu_0\Delta H$ = 5 T. The $\Delta S_M(T)$ curve at $\mu_0\Delta H$ = 5 T for the designed bed is displayed as dotted line in Figure 3a, the corresponding curves for the three wire components is shown in Figure 3b. The fluctuation of the magnetic entropy change for the designed arrays may be caused due to the $Gd_{60}Al_{20}Co_{20}$ component, which shows fluctuation at the transition region (Figure 3b). For exploring the relationship between the designed bed and the used wires, a simple model is used and it is assumed there are no interactions among different components, and then the magnetic entropy change in the designed bed based on the MCE of the used components is calculated by employing [31]:

$$\Delta S_M^{Design} = \alpha\Delta S_M^1 + \beta\Delta S_M^2 + \gamma\Delta S_M^3 + \cdots\cdots + \omega\Delta S_M^n \qquad (4)$$

where $\Delta S_M{}^{\text{Design}}$ is the magnetic entropy change in the designed microwire arrays. $\alpha$, $\beta$, $\gamma$ and $\Delta S_M{}^1$, $\Delta S_M{}^2$, $\Delta S_M{}^3$ are the weight fractions and magnetic entropy changes in the employed materials, respectively. The black line in Figure 3a is the calculated result for the designed bed and fits with the experimental data very well, which means that Equation (3) is an effective model in calculating the MCE performance of a designed structure using component materials with different MCE values.

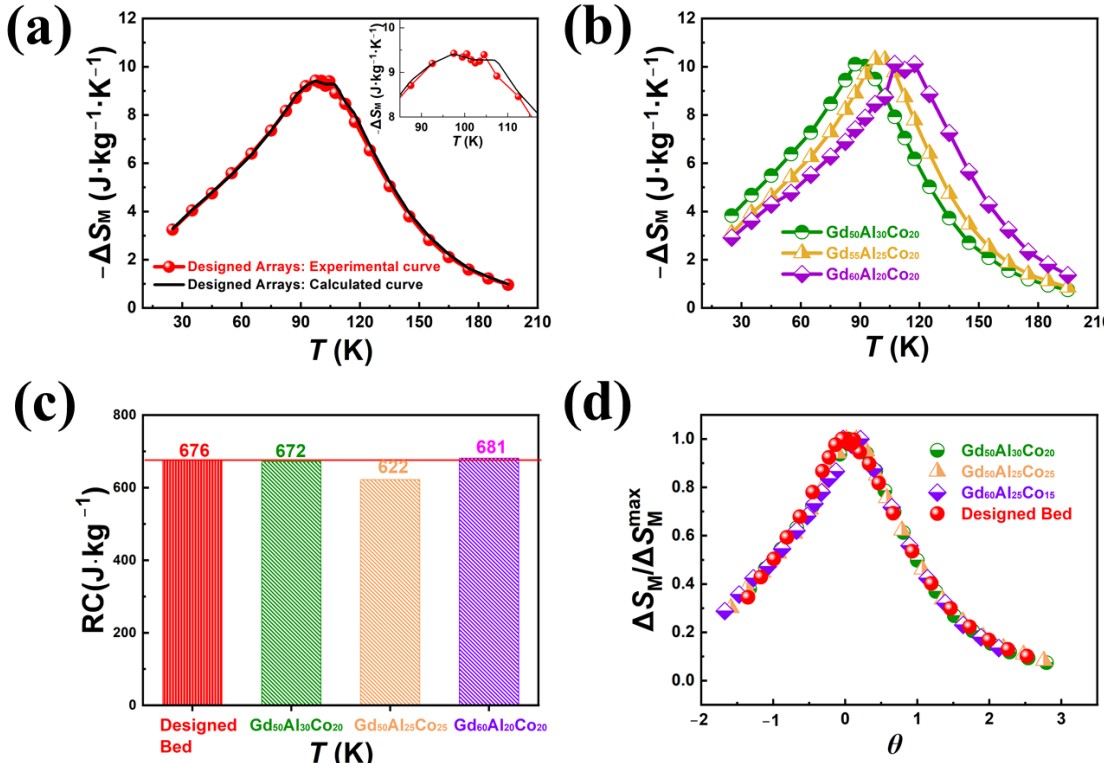

**Figure 3.** For $\mu_0\Delta H$ = 5 T. (**a**) The experimental data and calculated results of magnetic entropy change for the designed bed. (**b**) The magnetic entropy change curves of the used wire components. (**c**) The *RC* values and (**d**) universal master curves of designed bed and its Gd-Al-Co components.

The cooling efficiency of the designed microwire bed is calculated from Equation (3) and the value of refrigerant capacity (*RC*) is determined to be ~676 J/kg at $\mu_0\Delta H$=5 T. The *RC* values of $Gd_{50}Al_{30}Co_{20}$, $Gd_{55}Al_{25}Co_{25}$, and $Gd_{60}Al_{20}Co_{20}$ wires at $\mu_0\Delta H$ = 5 T are 672 J/kg, 622 J/kg, and 681 J/kg, respectively. The value of the designed bed is a little higher than ~659 J/kg, which is the average *RC* value of its three components (Figure 3c). This indicates that the cooling efficiency of the designed magnetic bed is not increased as much compared to its components. This is different with previous reports, which considered the smaller $T_C$ interval (10 K) of the used microwires.

For exploring the magnetic transition behavior of the designed bed and its difference with respect to the used Gd-Al-Co wire components, the universal master curves are established by collapsing all $\Delta S_M$ (T) curves at their external fields through normalizing their $-\Delta S_M{}^{max}$ values. The corresponding temperature is rescaled by the following equation [32]:

$$\theta = \begin{cases} -(T - T_C)/(T_{r2} - T_C) & T \leq T_C \\ (T - T_C)/(T_{r1} - T_C) & T \geq T_C \end{cases} \tag{5}$$

where $T_{r1}$ and $T_{r2}$ are two reference temperatures above and below $T_C$ which can be decided by the relation:

$$-\Delta S_m(T_{r1}) = -\Delta S_m(T_{r2}) = f \times (-\Delta S_m^{max}) \tag{6}$$

The value of $f$ is usually determined as 0.5 and the same is used in this work. The calculated results of the designed bed and its three components are plotted in Figure 3d. All the universal master curves at different external fields are fitted well for the used Gd-Al-Co microwires. The universal master curves at $\mu_0\Delta H$ = 5 T were chosen for comparison and all the designed beds and the used wire components are fitted very well, which explains why all the Gd-Al-Co wires show almost the same MCE and magnetic transition behaviors.

### 3.2. The Impact of $T_C$ Interval on MCE

For exploring the impact of $T_C$ interval of the used microwires on the MCE properties for the designed wire arrays bed, the virtualized data are employed based on the experimental data of the above three Gd-Al-Co wires. The magnetic entropy changes are immobile and only $T_C$ intervals are customized. Then, the magnetic entropy change curves and *RC* values of the designed arrays can be calculated from Equation (4). Based on previous reports [10], the *RC* value of the designed arrays increased when the used two components have a large $T_C$ interval. Thus, five wire arrays beds are designed (named D-1, D-2, D-3, D-4 and D-5) using two components with different $T_C$ intervals of ~10 K, 20 K, 30 K, 40 K and 50K. The magnetic entropy change curves of component A and component B are set the same as the experimental data of $Gd_{50}Al_{30}Co_{20}$ and $Gd_{50}Al_{25}Co_{25}$ microwires, respectively. The $T_C$ of component A is invariable (named A-Basic) and $T_C$ of component B is designed based on the required intervals (B-1, B-2, B-3, B-4 and B-5). The calculated results of $\Delta S_M(T)$ curves and *RC* values of the designed beds are shown in Figure 4a,b. Clearly, the designed beds exhibit increased table-like MCE regions with the increasing $T_C$ intervals, while the values of $-\Delta S_M{}^{max}$ decreased. Additionally, the *RC* values of the designed beds increased with the $T_C$ intervals, and the *RC* values are larger than that of the used two components when the $T_C$ interval exceeds 20 K.

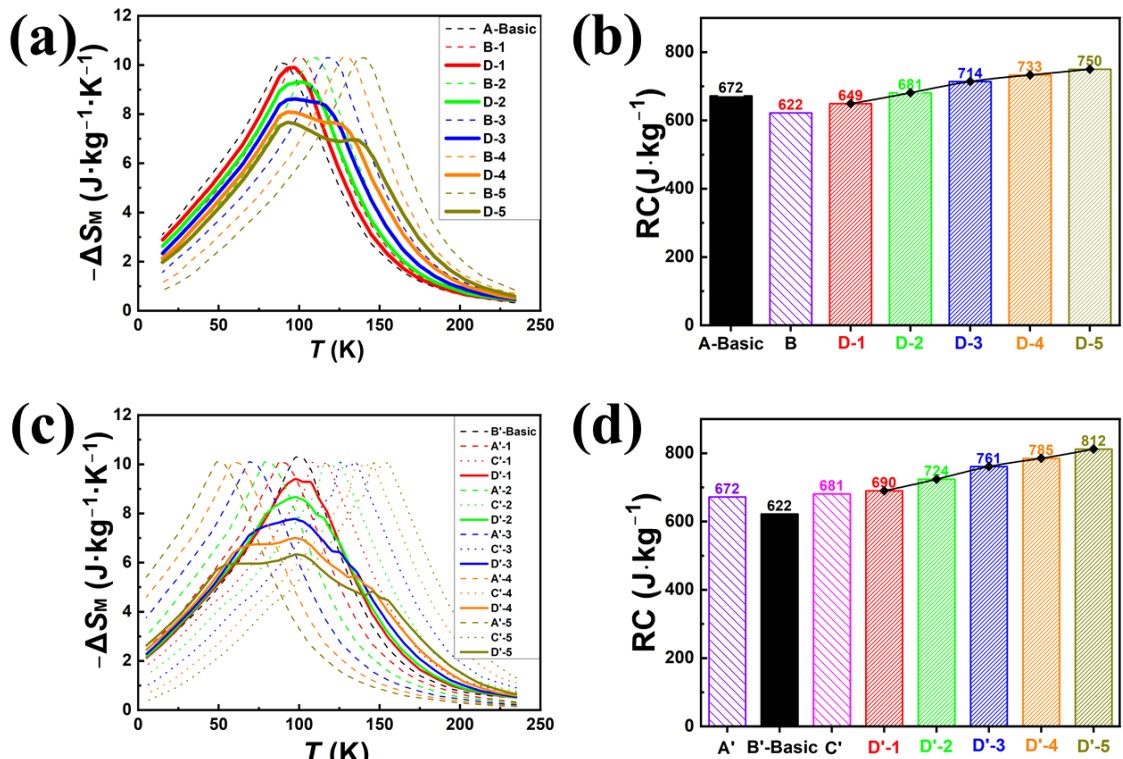

**Figure 4.** The calculated magnetic entropy change curves and *RC* values of the microwire-based beds with two components ((**a**) and (**b**)) and three components ((**c**) and (**d**)), respectively.

For the designed wire array beds with three components, five similar wire arrays beds are designed (named D'-1, D'-2, D'-3, D'-4 and D'-5) using three components with different

$T_C$ intervals of ~10 K, 20 K, 30 K, 40 K and 50K. The $\Delta S_M(T)$ curves of component A′, component B′ and component C′ are set the same as the experimental data of $Gd_{50}Al_{30}Co_{20}$, $Gd_{50}Al_{25}Co_{25}$, and $Gd_{60}Al_{20}Co_{20}$ microwires, respectively. The $T_C$ of component B′ is invariable (named B′-Basic) and $T_C$ of component A′ (A′-1, A′-2, A′-3, A′-4 and A′-5) and component C′ (C′-1, C′-2, C′-3, C′-4 and C′-5) are designed based on the required $T_C$ intervals. The calculated results of the magnetic entropy change curves and *RC* values of the designed beds are shown in Figure 4c,d. Similar to the case of the designed bed with two components, the designed beds with three components exhibit increased table-like MCE regions with the increasing $T_C$ intervals, while the values of $-\Delta S_M^{max}$ decreased. The *RC* values of the designed beds increased with the $T_C$ intervals, and the *RC* values are larger than that of the used three components, even when the $T_C$ interval is 20 K.

Notably, the microwires array beds with large table-like MCE ranges and large *RC* values can be easily designed using more magnetic wire components

## 4. Conclusions

In summary, the designed microwire array-based bed shows an excellent table-like MCE performance, ranging from ~92 K to ~107 K, and an enhanced cooling efficiency when using three kinds of Gd-Al-Co wire compositions. The designed bed achieved a ferromagnetic to paramagnetic transition with increasing temperature, and the average Curie temperature was determined to be ~100 K. The maximum magnetic entropy change $(-\Delta S_M^{max})$ and *RC* value of the designed bed were ~9.42 J/kgK and ~676 J/kg, respectively. The existence of the universal master curves clearly shows the designed bed with an MCE behavior like the used Gd-Al-Co components. Notably, the designed table-like MCE performance and enhanced *RC* can be achieved by using magnetic wire components with different $T_C$ intervals. These superior properties make the newly designed magnetic bed very promising for refrigeration in the liquid oxygen temperature range.

**Author Contributions:** Conceptualization, H.S. and L.L; methodology, H.S., L.L., H.B., S.J., L.Z. and J.L.; validation, H.S., Y.H., J.S. and M.-H.P.; formal analysis, H.S. and L.L. and H.B.; investigation, H.S., L.L., H.B. and M.-H.P.; resources, H.S., M.-H.P. and J.S.; data curation, H.S.; writing—original draft preparation, H.S., L.L. and H.B.; writing—review and editing, H.S., L.L., H.B. and M.-H.P.; visualization, H.S. and M.-H.P.; supervision, M.-H.P. and J.S.; project administration, M.-H.P. and J.S.; funding acquisition, H.S. and J.S. All authors have read and agreed to the published version of the manuscript.

**Funding:** This research was funded by Natural Science Foundation of Heilongjiang Province, grant number LH2019E054.

**Institutional Review Board Statement:** Not applicable.

**Informed Consent Statement:** Not applicable.

**Data Availability Statement:** The data presented in this study are available on request from the corresponding authors.

**Conflicts of Interest:** The authors declare no conflict of interest.

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
