# Peer review of "The Magnetocaloric Behaviors of Gd-based Microwire Arrays with Different Curie Temperatures"

_metals, doi:10.3390/met12091417_

Round 1
Reviewer 1 Report
good and interested work
Author Response
Thanks very much to the Reviewer comments.
Reviewer 2 Report
Manuscript Number: metals-1867981
Title: The Magnetocaloric behaviors of Gd-based Microwire Arrays 2 with Different Curie Temperatures
Review
The magnetocaloric effect (MCE), the basis for magnetic refrigeration, is a consolidated phenomenon in both theoretical and experimental aspects, with a large number of articles published in the areas of magnetism, magnetic materials, thermodynamics, and engineering materials. There are several works published in the literature on experimental investigations of the magnetocaloric effect observed in many magnetic systems. This number has increased considerably after the discovery of the giant magnetocaloric effect in the Gd_{5}Si_{2}Ge_{2} compound in 1997. In addition to classifying and qualifying materials in terms of their application to magnetic refrigeration, the calculation of magnetocaloric potentials enables the understanding of the microscopic mechanisms involved in magnetic phase transitions.
In this work, the authors carried out a study on the magnetic and thermodynamic properties of the Gd-based microwire arrays, in order to characterize its magnetocaloric potentials. Interesting how the table-like effect is obtained in this work. All data and results were very well presented and discussed. The cited references were well chosen and adequate. These materials present good magnetocaloric properties and are possible candidates for magnetic refrigerants with the application at low temperatures, for example, the liquefaction and storage of hydrogen gas.
I think that the topic of this study is quite interesting and that the results are in principle suitable for publication in Metals. However, there are open questions and deficiencies in the presentation that should be addressed before the publication of this manuscript can be recommended. These points are listed in the following.
-
Line 33. Is there a difference between the peak shape magnetic changes of crystalline and amorphous materials?
-
Line 85. Changes in magnetic entropy or total entropy?
-
Line 117. The decrease in the ΔS value of the bed is small but observable. The authors comment in the text that it was due to the Tc interval between the Gd-Al-Co wires. I don't understand how the magnetic transition affects this value. It is usually affected, in this case, by the replacement of a magnetic element by a non-magnetic one.
-
Equation 6. Will the f factor always be half of the peak? Or will the peak shape, higher or lower, affect your choice? Amorphous materials show more domed peaks in the magnetic transition, compared to their equivalent compound in crystalline form.
-
Figure 2. What is the direction of application of the magnetic field?
-
Section 3.2. The calculation is made only considering the impact of Tc on the caloric properties of the virtual systems assembled. For a real physical system, the variation in Tc is caused by the variation in the stoichiometry of a given alloy. But this variation causes a variation in the ΔS peak. Is not true?
-
Conclusions. What are the types of applications for the materials studied? The table-like effect was observed around the temperature of 100K. Far from the range where the usual refrigerators operate.

Author Response
Thanks very much. Please see the attachment. Please see the attachment.

Reviewer 3 Report
The paper by H. Shen et al. presents interesting and valuable experimental results and design procedures intended to obtain improved magnetocaloric properties based on microwire arrays. From the scientific point of view, the paper is of good quality and should be published. The minor questions are:
- How were the microwires handled to form a stack? The experimental method should be described in more detail.
- How was the magnetic field oriented in respect to the axis of the wires? Was there a difference between the results for different orientations? This question mainly concerns the effect of shape anisotropy.
Overall, the manusrcript provides novel data and analysis that can be of interest to the magnetocaloric society. However, I do not recommend publishing the paper until the English grammar has been thoroughly corrected, preferable by a native speaker. The text quality is very poor.
Author Response

(The authors gave the same response as above.)
